# Mapping a hidden terrane boundary in the mantle lithosphere with lamprophyres

Arjan H. Dijkstra [1] & Callum Hatch[1,2]

Lamprophyres represent hydrous alkaline mantle melts that are a unique source of information about the composition of continental lithosphere. Throughout southwest Britain, post-Variscan lamprophyres are (ultra)potassic with strong incompatible element enrichments. Here we show that they form two distinct groups in terms of their Sr and Nd isotopic compositions, occurring on either side of a postulated, hitherto unrecognized terrane boundary. Lamprophyres emplaced north of the boundary fall on the mantle array with $\varepsilon_{Nd}$ $-1$ to $+1.6$. Those south of the boundary are enriched in radiogenic Sr, have initial $\varepsilon_{Nd}$ values of $-0.3$ to $-3.5$, and are isotopically indistinguishable from similar-aged lamprophyres in Armorican massifs in Europe. We conclude that an Armorican terrane was juxtaposed against Avalonia well before the closure of the Variscan oceans and the formation of Pangea. The giant Cornubian Tin-Tungsten Ore Province and associated batholith can be accounted for by the fertility of Armorican lower crust and mantle lithosphere.

[1] School of Geography, Earth and Environmental Sciences, Plymouth University, Plymouth PL4 8AA, UK. [2] Department of Core Research Laboratories, Natural History Museum, Cromwell Road, London, SW7 5BD UK. Correspondence and requests for materials should be addressed to A.H.D. (email: arjan.dijkstra@plymouth.ac.uk)

Wilson's cycle[1] of the opening and closing of ocean basins throughout Earth history was based on the similarity of Early Palaeozoic faunal assemblages in the Avalon Peninsula of Newfoundland and in Southern Britain, which were strikingly different from fauna of the same age in the rest of North America and in Northern Britain. These faunas had evolved on either side of a 'Proto-Atlantic' ocean. This eventually led to the notion of an 'Avalonian terrane', whose northern margin is represented by the Caledonian suture with Laurentia[2]. Subsequently, structural geology, palaeomagnetism and geochronology have been key among the many disciplines in Earth Sciences used to map out and trace the movements of the many tectonic terranes from which present and past continents were pieced together[3–9]. It is now clear that Avalonia is one of a collection of peri-Gondwana terranes, lithospheric fragments that rifted away from Gondwana and were accreted to Laurentia throughout the Early Paleozoic[7,8,10,11]. Their movements as independent terranes ended with the Variscan Orogeny and the formation of the supercontinent Pangea, complete by the Late Carboniferous.

A key locality of Avalonia's southern margin is southwest Britain, where Avalonia is juxtaposed against another peri-Gondwana terrane, Armorica. Early Paleozoic faunas in southern Britain differ from those in Brittany, France, showing that Avalonia and Armorica where separated by an intervening ocean basin, the Rheic Ocean, in Silurian-Devonian times[2]. The Lizard Ophiolite, exposed on the southernmost edge of Britain, is widely considered to be one of the best-preserved fragment of the Rheic Ocean and the locus of the suture[12]. Different terrane analysis approaches have shown that Avalonia can be traced back to a position next to South-American Gondwana, while Armorica originated closer to the African part of Gondwana[3,5,13,14]. There are, however, problems with the interpretation of the southern margin of Avalonia in Britain. Post-Variscan lamprophyre dykes found throughout Armorica have a strong subduction-type geochemical signature[15,16], which is consistent with Armorica forming the overriding plate during the closure of the Rheic Ocean. However, identical igneous rocks in southwestern Britain —i.e., north of the Rheic suture—discussed in this paper, cannot be so easily explained if they are sited on the down-going Avalonian plate. Moreover, the Lizard Ophiolite has characteristics of a narrow Red Sea-like oceanic basin formed in a transtensional setting[17,18] rather than of a full ocean basin formed at a mid-ocean ridge. Recent revisions of Variscan tectonics in Europe have highlighted the role of many small ocean basins[19] and northward subduction of the Rheic Ocean and docking of Armorican terranes already in the Late Silurian, well before final closure of the other remaining oceanic basins in the Carboniferous[9]. If the Lizard Ophiolite was derived from one of these many small ocean basins, then it is possible that as yet unknown fragments of Armorica are present in southern Britain.

Lamprophyres are relatively rare volcanic or subvolcanic rocks characterized by dark mica or amphibole as the main phenocryst phase in a feldspar-rich groundmass[20]. The generally primitive nature of lamprophyres combined with high potassium and water content suggests that they are derived from previously metasomatised—probably veined—continental mantle lithosphere[20–28], and they are genetically linked to kimberlites, lamproites, carbonatites and ultramafic lamprophyres[20,21]. Therefore, they are a unique source of information about the composition of the deep parts of the continental lithosphere.

Post-orogenic calc-alkaline lamprophyres are relatively abundant in the Variscan Orogen of western and central Europe[15,16,28–31]. In addition, at least 30 localities of lamprophyres and closely related igneous rocks are known in southwestern Britain[32–35]. The lamprophyres are mica-rich and typically form 10 cm to m-wide dykes and other types of minor intrusions cutting across Variscan foliations in Carboniferous and Devonian rocks. Lamprophyre magmatism occurred between 295 and 285 Ma, coinciding with the first pulse of granite magmatism in the region[34]. Similar-aged potassic lavas are also found in the area, intercalated with Early Permian clastic sediments in graben structures[33]. Despite being largely mica-free, these calc-alkaline high-K lavas have trace element compositions showing that they are related to the lamprophyres[33–35].

This paper offers a new, mantle-based perspective on the tectonic make-up of the Avalonian margin: we use the compositions of lamprophyres to map distinct chemical domains in the mantle lithosphere of southwest Britain, revealing the presence of a hitherto unrecognized Armorican terrane fragment that lies hidden beneath Paleozoic rocks.

## Results

**Petrography.** The chemical compositions of samples of lamprophyres and potassic lavas from 22 locations in southwest Britain are reported in this paper (Fig. 1; see supplementary figure 1 and supplementary table 1 for details about locations). Many of the localities are poorly exposed overgrown quarries, but a few localities (e.g., MAW, PEN, and CRA) provide well-exposed lamprophyres in contact with country rocks (supplementary figure 2a). Nd and Sr isotope systematics are also reported for a subset of samples to better constrain the composition of their mantle source. Chemical compositions are reported in supplementary data 1.

The majority of samples are minettes consisting of phenocrysts of dark mica with <5% altered olivine phenocrysts in a finer-grained groundmass of feldspar, predominantly alkali-feldspar, mica, apatite and quartz. The mica phenocrysts generally consist of pale phlogopite cores with dark biotite rims (supplementary figure 2b). Two samples (CRA and LUR) are kersantites, containing dark mica in a plagioclase-dominant groundmass; the kersantites also contain relatively abundant chlorite. Habits of pseudomorphs after olivine are often blade-shaped or skeletal (supplementary figure 2c, d) in the lamprophyres. Many samples contain igneous carbonate (supplementary figure 2e). Most of the lamprophyres contain round mm-sized quartz xenocrysts (supplementary figure 2c), but other xenolithic material is uncommon. Some samples contain small autolithic inclusions of phlogopite-rich cumulate (supplementary figure 2f). Some dark mica phenocrysts have textures that seem reminiscent of sieve textures in their core, overgrown by contiguous rims of phlogopite grading outwards to biotite (supplementary figure 2b); these are interpreted here as remelted antecrysts or xenocrysts. Olivine is always strongly altered to carbonate or to serpentine; this is interpreted as largely due to autometasomatism[20] as a result of the high volatile content of the magmas, as completely altered olivine is found in otherwise completely fresh lamprophyres. Some clinopyroxene is found in rims around xenoliths of quartz-rich sediments. Two lamprophyric samples are vesiculated (WAS and HOL) and have an aphanitic groundmass, resembling lamprophyric lavas. Four samples of mica-free high-K lavas (DUN, KNO, POS and POC) are also included in the study. These lavas contain (altered) olivine and plagioclase phenocrysts in an aphanitic K-feldspar-rich groundmass.

Based on petrographic analysis, samples with predominantly clear, non-clouded feldspars, with primary carbonates and with clearly zoned micas with a pale centre and darker but clear pleochroic rim resembling biotite, are deemed 'fresh' in our study (e.g., supplementary figure 2b, e, f). Other samples are moderately altered, containing feldspars with cloudy patches, generally darker micas showing development of opaque rims and inclusions (e.g.,

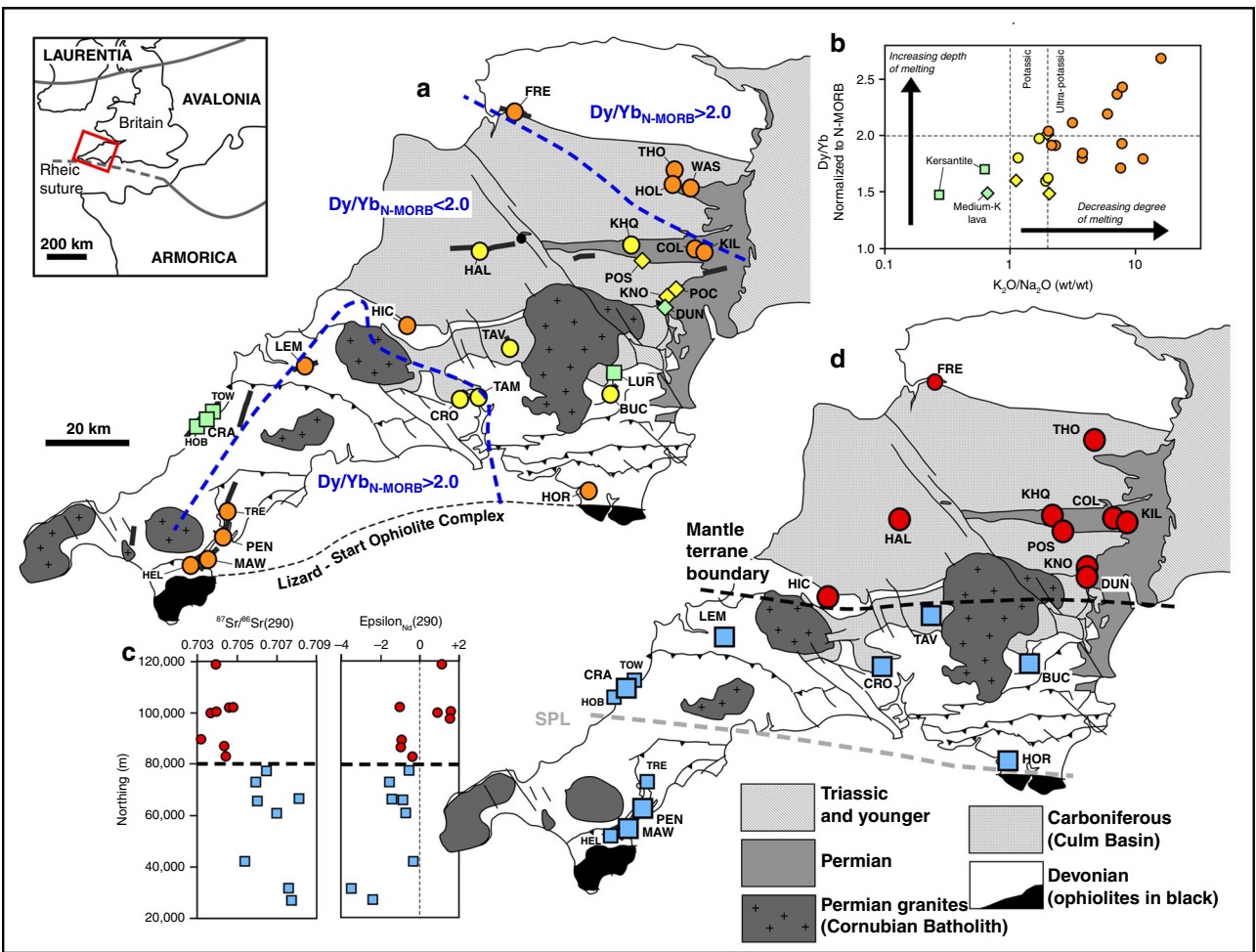

**Fig. 1** Geospatial analysis of post-Variscan lamprophyre geochemistry in southwest Britain. Inset map shows location of study area, with the generally assumed location of the Rheic suture marked by the Lizard-Start ophiolites complex. **a** Map of the study area with sample localities. Samples marked with circles are minette-type lamprophyres, squares are kersantite lamprophyres, and diamonds are mica-free K-rich lavas. Samples are also colour-coded based on $K_2O/Na_2O$ ratios. Orange samples are ultrapotassic ($K_2O/Na_2O$ >2.2) and represent the lowest degree of mantle melting; yellow symbols are potassic (1 < $K_2O/Na_2O$ <2.2) and green symbols represent samples with $K_2O/Na_2O$ <1. Two dashed contour lines delineate areas in north and south where deepest-derived magmas were emplaced, based on N-MORB normalized Dy/Yb ratios >2. **b** Chart showing negative correlation between depth and degree of melting. **c** Initial Sr and Nd isotope ratios calculated at 290 Ma plotted against northing (Ordnance Survey UK grid coordinates), showing a clear jump in values across thick dashed line. **d** Map showing the samples assigned to group 1 (red circles) and group 2 (blue squares) based on their initial Sr and Nd isotope ratios. Surface trace of the boundary between the two isotopically distinct lithospheric mantle domains is interpreted as a cryptic terrane boundary in the mantle lithosphere buried beneath Paleozoic metasedimentary rocks. SPL refers to the Start-Perranporth Line (see text). Data for locations TOW (Towan Head), HOB (Holywell Beach), TRE (Trelissick), HEL (Helfort) and FRE (Fremington Quay) are from ref [35]. (only Nd data); all other data from this study. Locations are listed in Supplementary Table 1. Map adapted from regional view geological map from British Geological Survey[59]. © Crown Copyright and Database Right 2018. Ordnance Survey (Digimap Licence)

supplementary figure 2c, d). Strongly altered lamprophyres are not included in our study, but K-rich basaltic lavas generally show significant alteration, with feldspars strongly altered to secondary saussurite assemblages, and with formation of abundant opaque phases.

**Chemical compositions**. The lamprophyres studied here are typical calc-alkaline lamprophyres: they generally have intermediate compositions ($SiO_2$ = 51–57%), high alkali and volatile content, and strong enrichments in large ion lithophile trace elements (LILE) when normalized to normal mid-ocean ridge basalt[36] (Fig. 2). The minette-type lamprophyres are highly potassic ($K_2O/Na_2O$ >1), and approximately half of the samples are classified as ultrapotassic ($K_2O/Na_2O$ > 2.2). The value of 2.2 instead of 2.0 is chosen as the lower limit, as these samples form a coherent group in the $K_2O$–$SiO_2$ diagram in supplementary figures 3a, b.

The relatively high $SiO_2$ and high alkali contents are coupled with primitive magma characteristics such as the presence of (altered) olivine phenocrysts, high bulk mg-numbers (mg# = molar $Mg/(Mg + Fe)$ up to 0.73) and high Ni and Cr (typically 100–200 and 150–600 ppm, respectively, supplementary figure 3c, d). The skeletal nature of the olivine observed in several samples excludes a xenocrystic origin. These primitive characteristics rule out an origin of the parental magmas by extensive crustal contamination, and they are regarded as near-primary mantle melts. Experiments show that in a hydrous mantle source, pyroxene makes a relatively large contribution to the partial melting reaction compared to olivine, leading to relatively $SiO_2$-rich (52–64 wt% $SiO_2$) primary magmas[37].

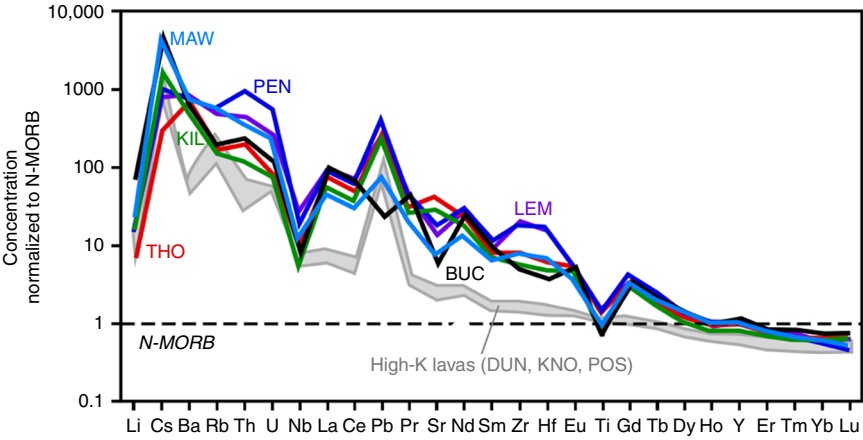

**Fig. 2** Extended trace element spidergrams for selected samples. Shown are six of the least altered lamprophyres, and three potassic calc-alkaline lavas, with concentrations normalized to normal mid-ocean ridge basalt (N-MORB). Diagrams show the enrichment in incompatible elements and the steep slope of the heavy rare earth elements

The lamprophyres are strongly enriched in incompatible elements, but depleted in Nb and Ti compared to other incompatible elements, and depleted in the heaviest rare earth elements (REE) Er to Lu compared to normal mid-ocean ridge basalt (N-MORB) (Fig. 2). A relatively deep origin for the parental magmas is confirmed by the steep heavy REE (HREE) slopes (Yb/Dy normalized to N-MORB fall in the range 1.5–2.6), generally interpreted as the signature of residual garnet in the source[38]. This indicates a depth of origin >60–85 km[38], with the highest ratios indicating a source close to the base of the lithosphere at the time, assumed to be at least similar to the present-day depth of the lithosphere–asthenosphere boundary of 100–125 km in southern Britain[39].

**Sr and Nd isotopes.** While lamprophyres from across the study area are broadly similar in mineralogy, texture and bulk major and trace element composition, their Nd and Sr isotopic compositions fall into two clearly distinguishable groups (Fig. 3). One group of samples of lamprophyres exhibit initial isotopic compositions that coincide with the mantle array line between Bulk Silicate Earth and Depleted Mantle for 290 Ma, with initial $\varepsilon_{Nd}$ values of −1 to +1.6. The potassic lavas studied also fall in this group; their Nd isotopic signature shows that they formed from lithospheric magmas closely related to the lamprophyres, and that they do not have a petrogenetic affinity with shoshonitic arc magmas[26]. The second group of samples shows a displacement off the mantle array line to higher, more radiogenic initial $^{87}Sr/^{86}Sr$ ratios, with negative initial $\varepsilon_{Nd}$ values (−0.3 to −3.5).

## Discussion

The alkali and light REE (LREE) concentrations are used here as a proxy for the overall degree of melting in the mantle source: in a metasomatised source, mineral assemblages in mineral pockets or vein assemblages rich in LILE, LREE and volatiles have lower solidus temperatures compared to ambient mantle peridotite and will be the dominant contribution to a mantle melt at very low degrees of melting, whereas typical depleted mantle wall rocks will contribute progressively more to the magma as melting progresses[23,25]. Due to the continuous breakdown of hydrous minerals in the mantle during the melting, water is continuously available and the source remains fusible and can produce up to 20% melt[37]. In southwest Britain, ultrapotassic lamprophyres occur to the north and south of a central area of slightly elevated

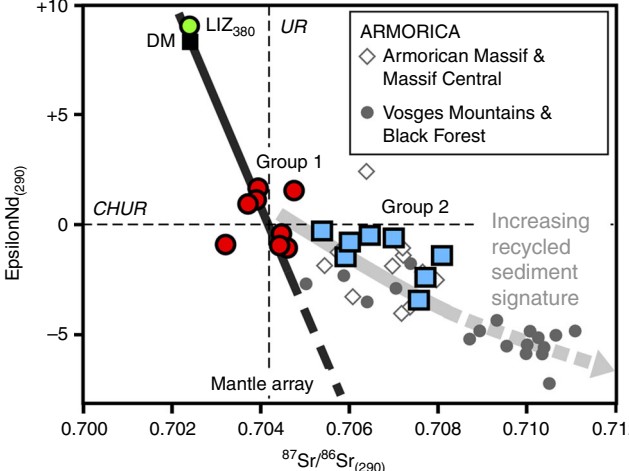

**Fig. 3** Initial Sr–Nd isotopic compositions of magmas from southwest Britain at 290 Ma. Group 1 samples (red circles) fall on the mantle array, whereas group 2 samples (blue squares) are systematically displaced to more radiogenic Sr isotopic values. Isotopic compositions for similar-aged lamprophyres from Armorican massifs in Europe (locations in Fig. 5; published data[15,16]) are shown for comparison (grey diamonds: lamprophyres from various Armorican massifs and the Massif Central in France; filled grey circles: lamprophyres from the Vosges mountains and the Black Forest). Group 2 lamprophyres from southwest Britain are isotopically identical to lamprophyres from Armorica. Green circle marks the composition of a 380 Ma olivine dolerite dyke from Coverack in the Lizard Ophiolite (LIZ$_{380}$) analysed alongside the lamprophyres. DM depleted mantle, CHUR chrondritic uniform reservoir (Nd), UR uniform reservoir (Sr)

degrees of mantle melting that yielded potassic lamprophyres and lavas with $1 < K_2O/Na_2O < 2.2$ (Fig. 1a). One lava (DUN) in this area of elevated melting is a high-K calc-alkaline basalt with $K_2O = 2.1$ wt% and $K_2O/Na_2O = 0.7$, representing the highest degree of melting. Similarly, the HREE slope Yb/Dy (garnet signature[38]) is taken as an indication of the relative depth of the source of the magma. There is an inverse correlation between degree and depth of melting in the area, with the most alkaline and LREE-enriched samples having the deepest origin (Fig. 1b). The mantle domain that experienced the shallowest and highest degree of post-orogenic, Early Permian mantle melting thus

mapped out (Fig. 1a, b) underlies a region of Carboniferous sedimentation (the Culm basin); this pattern is most easily explained as an area of localized lithospheric thinning causing low-degree decompression melting driven by Early Permian post-Variscan extension. The coincidence with the Carboniferous sedimentary basin suggests that the formation of this 'lithospheric neck' was already initiated during an Early Carboniferous phase of intra-plate extension.

Group 2 lamprophyres plot off the mantle array for 290 Ma towards more radiogenic Sr isotopic ratios, coupled with mildly lower $\varepsilon_{Nd}$ values. Alteration can be ruled out as a cause for this radiogenic Sr enrichment, as this group includes several very fresh samples (PEN, MAW and LEM). We investigated whether the radiogenic Sr isotopic compositions of group 2 samples can be explained by contamination of mantle-derived lamprophyric magmas with crustal material during the emplacement of the lamprophyres. Supplementary figure 4 shows the results of a test consisting of three mixing models between a typical group 1 lamprophyre magma (composition of KIL6) with three different contaminants. None of the models shown can plausibly explain the composition of group 2 lamprophyres by a contamination and assimilation process, as they require in excess of 35% crustal contaminant. Such high degrees of assimilation are wholly inconsistent with the primitive nature of many of the lamprophyres. Instead, the isotopic composition must reflect the mantle source. Rather than being exceptional, post-orogenic lamprophyres with radiogenic Sr isotope ratios are the norm in the Variscan belt of Europe, and have been recorded as far east as Poland[29]. In many recent studies, such Sr isotopic compositions in lamprophyres were interpreted as reflecting the isotopic signature of old subducted sediments in the mantle source, imparted by fluids derived from a subducting slab just before or during lamprophyre emplacement event[15,16,31,40]. Below we argue, however, that this signature in the mantle source of group 2 lamprophyres of southwest Britain may be the result of older, possibly Neoproterozoic–Cambrian metasomatism.

A significant discovery of this study is the spatial distribution of isotopic groups 1 and 2 lamprophyres: group 1 lamprophyres are only found in the north of the area while group 2 lamprophyres are only found in the south (Fig. 1c, d). The linear character and the perfect separation suggest that there is a steep boundary in the mantle lithosphere of southwest Britain. The strong Sr and Nd isotopic contrast between the domains on either side clearly indicates a long-term compositional difference and provides strong evidence for the presence of an ancient (Lower Paleozoic) steep terrane boundary that was hitherto unrecognized. A geochemical study using lamprophyres showed a clear Nd isotopic contrast in the mantle on either side of the well-exposed lithospheric-scale Great Glen Fault in Scotland[41]. The mapped mantle boundary in southwest Britain is, however, cryptic and does not have an obvious tectonic surface expression. It is parallel to several steep east–west faults recognized as having caused sedimentary basin segmentation in the Devonian sequences[42] of which one, the Start-Perranporth Line (SPL, Fig. 1d), was previously proposed as a crustal terrane boundary[43]. It is proposed here that these faults in the crust are near-surface splays of the much deeper lithospheric-scale transcurrent fault mapped here by lamprophyre isotopic compositions. The terrane boundary is overlain by the Carboniferous Culm Basin, and its surface trace is apparently 'stitched' by the Early Permian Dartmoor granite intrusion. Significant facies differences between Devonian sedimentary sequences on either side[44] are permissive of completion of terrane juxtaposition as late as the Middle or Late Devonian, but more likely represent the control of basement faults related to reactivation of the terrane boundary. The absence of obvious unconformities in the Devonian sedimentary

successions[44,45] suggests that the terrane boundary was formed not later than the Early Devonian. This seems to be broadly coeval with the postulated soft collision between the Armorican-derived terranes and Avalonia further in north-central Europe at the end of the Silurian[9].

The enrichment in radiogenic Sr of group 2 lamprophyres, interpreted as a subducted sediment signature, is absent in group 1 lamprophyres, which otherwise exhibit the same evidence for extensive potassic and volatile metasomatism in their mantle source. This suggests that the radiogenic Sr enrichment and the potassic metasomatism are two separate events. The former event is only found in the southern terrane and predates the terrane juxtaposition, while the latter affected the whole region, and thus postdates the terrane boundary (and is an 'overlap assemblage' in terrane analysis terminology).

In this case, the radiogenic Sr signature is not due to subduction of old sediment during lamprophyre emplacement[15,16,31,40], but resulted from partial melting of mantle lithosphere that had been modified by metasomatism in the past, prior to the terrane juxtaposition. The metasomatism probably involved sediment-derived fluids and formation of mica-peridotites. The lamprophyres of the southern terrane exhibiting the radiogenic Sr isotope signature are isotopically indistinguishable from similar-aged lamprophyres in Armorican massifs in Europe (Fig. 3). Given that the radiogenic Sr enrichment is so prevalent in the source of post-Variscan lamprophyres throughout Europe, the likely geological context for this event is the Cadomian Orogeny. This period of accretionary mountain-building at the active margin of Gondwana during the late Neoproterozoic to Cambrian has affected all major pre-Variscan continental blocks in Armorican Europe, which have otherwise disparate older histories[14,46]. The more widespread potassic-hydrous metasomatism that overprinted the terrane boundary can most easily be explained as having occurred above a north-dipping slab during Variscan subduction of oceanic lithosphere, although Late Devonian-Early Carboniferous alkaline intra-plate magmatism in the region[47] is not fully discounted here as a contributing cause of the metasomatism.

Seismic imaging of steep lithosphere-scale continental strike-slip zones in the mantle remains inherently difficult[48,49], and the lack of sharp Moho off-sets on major continental strike-slip zones is often explained by a distributed nature of the deformation in the lower crust and mantle[50,51]. This study shows that geochemical mapping of terrane boundaries using post-orogenic, lithosphere-derived igneous rocks such as lamprophyres can be a powerful complement to traditional geophysical methods. Our geochemical mapping of the base of the mantle lithosphere (>60–85 km) of southwest Britain has revealed the presence of a narrowly defined terrane boundary with an apparent width <20 km, with the terranes of either side having distinct isotopic compositions (Fig. 4). The terrane boundary can be tentatively correlated with a system of major transcurrent faults in Europe (Fig. 5).

Critically, since the lamprophyres of the southern terrane are isotopically indistinguishable from similar-aged lamprophyres in Armorican massifs in Europe, we conclude that the newly recognized terrane boundary juxtaposed Armorican mantle in the south against Avalonian mantle in the north. The implications of this conclusion are manifold. The southern margin of Avalonia in Britain is not defined by a single collisional suture, but instead by one or more steep transcurrent[45] terrane boundaries. The 'suture' defined by the Lizard Ophiolite is instead a structure related to the closure of a minor tract of the Rheic Ocean (Fig. 4). This is fully consistent with recent interpretations of the Lizard Ophiolite in a relatively small transtensional ocean basin[17]. Docking of Armorican fragments started well before the peak of the Variscan

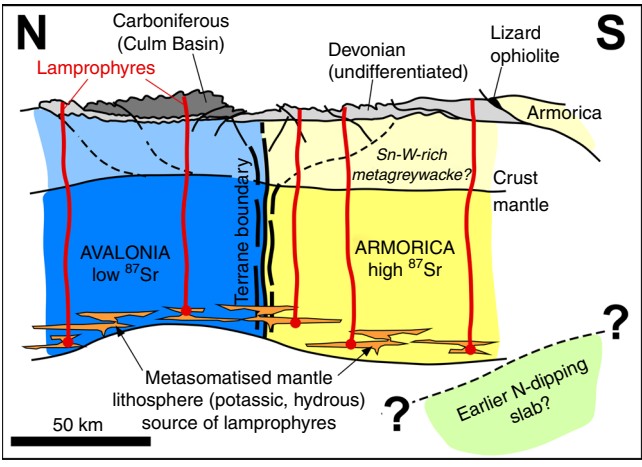

**Fig. 4** Schematic north–south cross-section showing the terrane boundary around the time of emplacement of the lamprophyres (c. 290 Ma), after the Variscan Orogeny. Armorican mantle lithosphere in yellow with characteristic high $^{87}Sr/^{86}Sr$ is juxtaposed against Avalonian mantle lithosphere in blue. Both domains had been affected by potassic-hydrous metasomatism (orange veins in basal part of the lithosphere), possibly above a (north-dipping?) subduction zone. These metasomatised mantle rocks formed the source for the lamprophyres on both sides. Deepest-derived lamprophyric magmas are formed on either side of a central area of thinned lithosphere. Dashed lines in lower crust denote inferred basement faults which controlled the segmentation of the Devonian sedimentary basins and which were re-activated as thrusts during the Variscan Orogeny. Upper crustal rocks (after published cross-section[42]) mainly comprised of Devonian and Carboniferous sedimentary rocks were probably deposited after terrane juxtaposition and are shown in grey

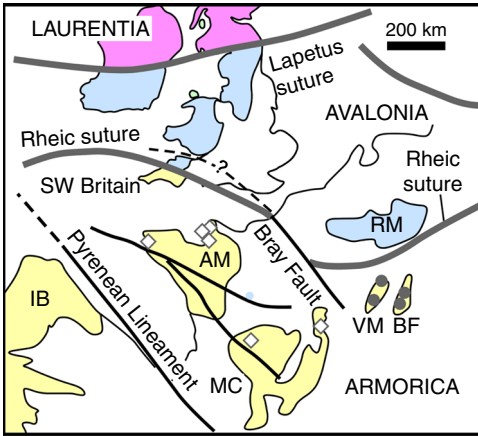

**Fig. 5** Location of the newly recognized terrane boundary in its wider tectonic context. Other major faults including the Bray fault and its previously proposed extension (dashed) in Britain[45]. Symbols show locations of the Armorican lamprophyres for which isotopic compositions are shown in Fig. 3 (using the same symbols). Pre-Variscan Massifs: AM Armorican Massif, MC Massif Central, IB Iberia, VM Vosges Mountains, BF Black Forest, RM Rhenish Massif. Map adapted from published tectonic map[45]

Orogeny, and the terrane juxtaposition in southwest Britain cannot be assigned unambiguously to either the Caledonian or the Variscan Orogeny. Ultimately, this shows that in Britain, just like in North America[11] and in Northern Europe[9,52], the closure of Wilson's (1966) 'Proto-Atlantic Ocean' consisted of a

protracted history of accretion of terranes, rather than two (Caledonian and Variscan) punctuated collisions events.

Finally, the post-Variscan giant Cornubian Sn-W orefield and the associated peraluminous granitic batholith of southwest Britain[53] are superimposed on the Armorican terrane, and the general absence of mineralized veins north of the terrane boundary is striking (Fig. 6). Similar mineralization associated with peraluminous granitoids can be found throughout the Armorican massifs of Europe, most notably in the giant Erzge-birge Ore Province[54–56]. This shows that the Armorican lower crust generally had the right composition (e.g., metagreywacke[54]) to produce the Sn-W-rich peraluminous granitic magmas, as opposed to the crust of the Avalonian terrane. The lamprophyre magmas transferred fluids as well as heat-producing elements (K and Th) from the metasomatised lithospheric mantle to the crust, and thus probably played a significant role in crustal melting and formation of the mineral resources.

## Methods

**X-ray fluorescence (XRF).** All XRF samples were prepared and analysed in the Plymouth University ISO9001:2008-certified Consolidated Radioisotope Facility (CoRiF). Milled sample material (<50 µm) and flux (67% $Li_2B_4O_7$: 33% $LiBO_2$) were oven dried overnight at 105 °C prior to fusion using an Eagon 2 furnace (PANalytical). Sample and flux were mixed at a ratio of 0.9 g sample to 9 g flux in Pt–Au crucibles (95% Pt; 5% Au) and fused at 1200 °C for 10 min. The samples were cast into glass discs using 40 mm Pt–Au dishes with $NH_4I$ added as a releasing agent prior to casting. In addition to the sample material, reference materials (JB-2 basalt, OU-5 Leaton dolerite, BE-N alkali-basalt and OKUM komatiite) of similar matrix were prepared in the same manner to validate the procedure. Repeatability was assessed by preparing several samples in triplicate. Loss on ignition (LOI) was determined on separate subsamples of the dried material by igniting at 1050 °C for 1 h in a muffle furnace. A sample of known LOI was included in each batch following internal quality-control procedures. Samples were analysed for 17 ele-ment oxides using a WD-XRF spectrometer with gas flow and scintillation detectors (Axios Max, PANalytical). The instrument was operated at 4 kW using a Rh target X-ray tube. During sequential analysis of elements, tube settings ranged from 25 kV, 160 mA for low atomic weight elements up to 60 kV, 66 mA for higher atomic weight elements. The instrument was calibrated using synthetic calibration standards and measurement conditions were optimized using SuperQ software and the WROXI analysis application (PANalytical). Instrument drift was assessed following internal quality-control procedures using a multi-element glass sample. Accuracies based on comparison with certified reference materials reported as % error are <2.5% for major element oxides, $TiO_2$, MnO, $K_2O$ and $Na_2O$, and <5% for Cr and Ni in the relevant concentration range. Lower limits of detection for the samples were <50 ppm for most elements, and 27–29 ppm and 13–15 ppm for Cr and Ni, respectively.

**Trace element analysis.** Trace elements were measured by inductively coupled plasma mass spectrometry (ICP-MS) using a Thermo Fisher X-Series II at the University of Southampton, National Oceanography Centre Southampton, fol-lowing an $HNO_3$–HF digestion. Samples and standards were spiked with internal standard elements and corrected for interferences and blank and then calibrated using a suite of international rock standards (JB-3, JB-1a, BHVO-2, BIR-1 and JA-2) and in-house reference materials. Long-term accuracy relative to reference values is 3–5% for all trace elements.

**Sr and Nd isotope analysis.** Sr was separated for isotopic analysis by column chemistry using Sr-spec resin. Sr isotopes were analysed on a Thermo Scientific Triton Thermal Ionization Mass Spectrometer at the University of Southampton, National Oceanography Centre, Southampton, using a static procedure with amplifier rotation (on an $^{88}Sr$ beam of >2 V). Fractionation was corrected using an exponential correction normalized to $^{86}Sr/^{88}Sr = 0.1194$. NIST 987 was run as a reference standard and the long-term average (>70 analyses) on this instrument is 0.710249 ± 0.000022 (2 SD).

Nd isotopes were measured at the University of Southampton, National Oceanography Centre Southampton, using a multi-collector inductively coupled plasma mass spectrometer (MC-ICP-MS, Thermo Scientific Neptune). The Nd fraction was purified from a whole rock digest via ion exchange column chemistry. Corrected Nd isotopic compositions were obtained through adjustment to a $^{146}Nd/^{144}Nd$ ratio of 0.7219 and a secondary normalization to $^{142}Nd/^{144}Nd = 1.141876$[57]. Results for the JNdi-1 reference standard[58] measured as an unknown were 0.512115 with an external reproducibility of the ±0.000006 (2 SD) across six analysis sessions over 2 years.

Comparison of our isotope geochemical results with those of previous studies in the area[33,35] shows that age-corrected (290 Ma) $^{143}Nd/^{144}Nd$ ratios for samples

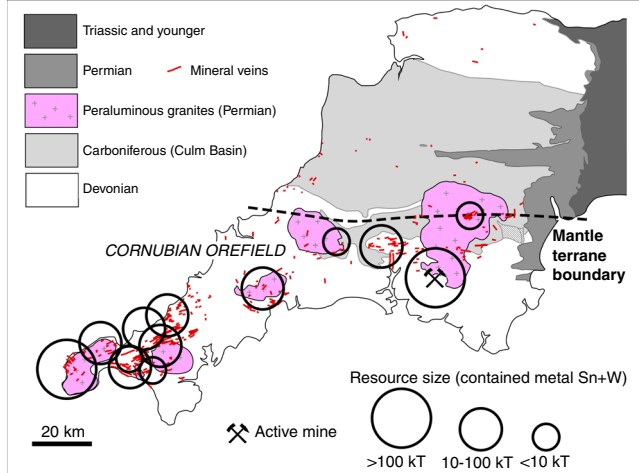

**Fig. 6** The distribution of mineral veins in southwest Britain and the location of the giant Cornubian W-Sn orefield. Estimated resource sizes (contained metal Sn and W in kilo-tonnes of reserves and resources) of the main ore districts recognized in the region are indicated by open circles (diameter proportional to resource size), after published data[56]. Also shown is the active world-class Hemerdon W-Sn mine. Mineral veins from 1:50,000 digital data set (DigimapGB-50) from British Geological Survey. Map adapted from regional view geological map from British Geological Survey[59]. © Crown Copyright and Database Right 2018. Ordnance Survey (Digimap Licence)

taken from the same localities (KIL, DUN, MAW, PEN, KHQ and POS) as reported by these authors were very well reproduced in our study, with relative deviations <0.007%. A similar comparison with published results for age-corrected $^{87}Sr/^{86}Sr$ isotope ratios for samples from localities KIL and DUN[33] yielded relative deviations of 0.024% and 0.002%, respectively.

Compositions of the depleted mantle (DM), chondritic uniform reservoir (CHUR), uniform reservoir (UR) and the 'mantle array' line connecting these reservoirs in Fig. 3 at 290 Ma were calculated using commonly used values: DM ($^{143}Nd/^{144}Nd = 0.513114$, $^{147}Sm/^{144}Nd = 0.2220$, $^{87}Sr/^{86}Sr = 0.7025$ and $^{87}Rb/^{88}Sr = 0.022$); CHUR ($^{143}Nd/^{144}Nd = 0.512638$ and $^{147}Sm/^{144}Nd = 0.1967$); and UR ($^{87}Sr/^{86}Sr = 0.7045$ and $^{87}Rb/^{86}Sr = 0.0827$).

### Data availability

The authors declare that all analytical data supporting the findings of this study are available within the paper and its supplementary information files. Fragments of samples discussed in this paper have been deposited at the Plymouth Museum, Galleries, Archives, and museum numbers are listed in Supplementary Table 1.

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

## Acknowledgements

This work is supported by funding from an FP7 Marie Curie Career Integration Grant (Os.Earth). Kevin Page and Richard Scrivener are thanked for their help in collecting samples. Various landowners are thanked for their permission to collect samples. Critical technical support by Rob Hall, Alex Taylor and Andy Milton is gratefully acknowledged.

## Author contributions

A.D. designed the project and carried out the analysis of the majority of the samples. C. H. collected and analysed the first 6 samples and came up with preliminary conclusions. Both authors contributed to the interpretation of the complete data set. The manuscript was written by A.D. with a significant contribution by C.H.

## Additional information

**Competing interests:** The authors declare no competing interests.

