## [Peer Review File · Nature Communications]

Reviewers' comments:

Reviewer #1 (Remarks to the Author):

In this well written manuscript (MS), the authors identify two isotopically and spatially discrete groups of Permian-age, post-collisional lamprophyres in SW England. It is suggested that the distribution of these lamprophyres can be used to map out a hitherto unidentified lithospheric terrane boundary between Avalonia and Armorica. These conclusions differ markedly from most previous work that places this terrane boundary further to the south through the Lizard peninsula.

I have no major reservations about the data, the science or the interpretation offered by the authors and I think their findings make a valuable and novel contribution to our understanding of the pre-Variscan history of SW England. However, I have three main quibbles that I think need addressing:

1. Based on the criteria against which I have been asked to review this MS, I would question whether the method of using lamprophyres/magma chemistry to map terrane boundaries is a truly novel approach. For example, a very similar approach was used to map out many of the tectonic terranes across Scotland in the 1980s (e.g. Thirlwall, 1982).

2. Can the authors demonstrate that this study has important implications that extend beyond SW England? In other words, can the authors demonstrate that these findings will be of wide interest to those who do not work on the geology of SW England? If so, then I think the MS could be 'better sold' to appeal to a wider audience.

3. Is there any seismic evidence to support the idea that a steep-angled terrane boundary exists beneath the Culm Basin?

Although points one and two will ultimately depend on the Editor's discretion, I would encourage giving the authors an opportunity to address them.

In addition to the points above, I have included a number of more minor comments/suggestions below:

Lines 6-8: See point 1 above. Have the authors overstated how novel the approach is given the use of similar techniques to map out the Caledonian terranes of Scotland? Again, I should stress that I have no reservations about the novelty of the findings.

Line 11: I think it would be fairer to stress that this is a 'supposed' lithospheric discontinuity.

Line 15: "subtle surface expression" seems rather vague. Is it simply the case that there is no surface expression?

Line 34: that instead of which

Line 36: change the expression "their wandering years"

Line 37: Give the age of the Variscan Orogeny for less familiar readers

Line 50: Are most ophiolites not considered rather anomalous (tectonically and compositionally) by comparison to standard MORs?

Line 52: Could metasomatism have occurred even earlier?

Line 60: See previous comment about how novel the method is.

Line 84-87: Are you referring to published data about the similarly aged potassic lavas? If so, a reference is needed.

Line 94-97: I would include the names of the labs used here. I appreciate that this is included in the supplementary information.

Line 103 – What exactly is meant by 'high alkali and volatile contents'?

Line 119 – reference needed for 60 km depth comment

Line 120 – reference for lithospheric thickness in this region?

Line 134 – I'm not really sure what the term 'melting anomaly' refers to. Can this be changed/clarified?

Line 135 – proxy rather than measure

Line 148 – other studies use similar trace element ratios to identify garnet signatures. Please reference

Line 150 – again, please clarify what you mean by 'melting anomaly'?

Line 160 – The argument for these samples not being altered could do with strengthening. Simply saying that the samples look fresh is not sufficient. Can some form of cryptic alteration be ruled out?

Line 162 – use colon after 'well'

Line 172 – why can't metasomatism have occurred much earlier?

Line 181 – change 'is' to 'provides'

Lines 191-198 – there are some interesting points here that could be further developed (although I accept that space is limited). The absence of any unconformity within the Devonian sequence of SW England is in marked contrast to the Devonian successions further north in Wales and northern England which are punctuated by the c. 400 Ma Acadian unconformity. SW England therefore appears to have escaped Acadian deformation. Woodcock et al (2007) highlighted these issues and suggested that SW England may have been juxtaposed with the Acadian belt along the Bristol Channel Fault Zone during the latter part of the Variscan. These authors favour a model where SW England was located adjacent to the Brabant Massif (Belgium) during the Devonian, thus allowing it to escape Acadian deformation. Interestingly, an earlier (early Devonian) deformation even (the Brabantian) is recognised in the Brabantian Massif that pre-dates the Devonian successions of SW England. Might the amalgamation of the two terranes proposed in this MS be related to the Brabantian event if the model of Woodcock et al (2007) is accepted?

Line 227 – same question about how novel the method is.

Line 228: again, any seismic evidence to support the idea that a steep terrane boundary exists?

Line 239 – delete 'already'

Line 245-254 – Whilst I accept that it seems likely that mineralisation may be a characteristic of the Armorican terrane, the absence of any granites in the Avalonian terrane means that it's potential for generating mineralised ore fields is limited.

Figures – these seem fine. I would consider re-ordering some of the figures such that a map is presented early on.

I wish the authors all the best in addressing these comments and I look forward to reading their published work

Yours Sincerely
Andrew Miles
30th April 2018

Reviewer #2 (Remarks to the Author):

Mapping a hidden terrane boundary 1 in the mantle lithosphere with lamprophyres. Review by Brendan Murphy

A plethora of tectonic studies inevitably focus on the definition and timing of crustal boundaries (sutures) between terranes. But when crustal fragments collide, the fate of their respective sub-continental lithospheric mantle (SCLM) is rarely considered. This is a well written manuscript that uses geochemical and isotopic tracking to deduce the location of boundary between (respectively) sub-continental lithospheric mantle that lies beneath Avalonia and Armorica (Cadmia).

Isotopic tracking the evolution of the SCLM has been done before. The innovation here is that the approach is employed to deduce the deep mantle expression of a terrane boundary and that lamprophyres are to compare the deep (garnet lherzolite) SCLM beneath both terranes.

The results would be of interest to the many geologists working in southern Britain, and more generally to a global set of geologist who are trying to employ innovative methods to deduce the existence of terrane boundaries.

I have made many comments directly on the annotated pdf (attached). **[Editorial Note: Due to journal policy, an annotated pdf cannot be published as part of the Peer Review File.]** Most of these comments deal with clarifying the text, or some judicious rephrasing rather than scientific issues. Accordingly, I think this manuscript is acceptable after minor revisions. The authors should evaluate (rather than accept) the suggested edits, because some of them may have inadvertently changed the meaning they intended.

My main concern is the rather cavalier way in which the potential effects of alteration are treated. I have seen these rocks in the field and in thin section and they ARE altered. This alteration is unlikely to significantly affect the more robust Sm-Nd isotopic systematics, but they are certainly likely to affect the more fickle Rb-Sr isotopic systematics. The differences in Sr isotopic signature seem more profound than Nd. Therefore the author(s) should justify their assumption about how representative the Sr isotopic analyses are of the original magma composition.

Reviewers' comments (in black) and our replies (in blue).

Reviewer #1 (Remarks to the Author):

In this well written manuscript (MS), the authors identify two isotopically and spatially discrete groups of Permian-age, post-collisional lamprophyres in SW England. It is suggested that the distribution of these lamprophyres can be used to map out a hitherto unidentified lithospheric terrane boundary between Avalonia and Armorica. These conclusions differ markedly from most previous work that places this terrane boundary further to the south through the Lizard peninsula.

I have no major reservations about the data, the science or the interpretation offered by the authors and I think their findings make a valuable and novel contribution to our understanding of the pre-Variscan history of SW England. However, I have three main quibbles that I think need addressing:

1. Based on the criteria against which I have been asked to review this MS, I would question whether the method of using lamprophyres/magma chemistry to map terrane boundaries is a truly novel approach. For example, a very similar approach was used to map out many of the tectonic terranes across Scotland in the 1980s (e.g. Thirlwall, 1982).

REPLY: We don't think that that is a completely accurate assessment. We repeat the words from reviewer 2 here: "Isotopic tracking the evolution of the SCLM has been done before. The innovation here is that the approach is employed to deduce the deep mantle expression of a terrane boundary and that lamprophyres are [used] to compare the deep (garnet lherzolite) SCLM beneath both terranes."

Our study is also different to other studies focussed on mantle heterogeneity in the sense that our sample set is deliberately compositionally highly restricted and that we therefore rigorously compare 'like with like', i.e., we mostly compare minettes with minettes. The parental magmas of these rocks can clearly be shown to be derived from the lithosphere (most mantle heterogeneity studies deal with the convecting part of the mantle), so we also compare like with like in terms of a very specific mantle source.

This is quite a difference to the approach by Thirlwall (EPSL, 1982) who mapped variations in mantle composition across the Caledonian arc - or the later paper by Thirlwall (Journal of the Geological Society, 1989), which is more explicitly about terrane boundaries. Whilst these studies were seminal in their nature, they use a range of rock types with variable sources (including asthenospheric ones), and are therefore not focussed on really mapping a distinct lithospheric mantle discontinuity.

Moreover, our sampling is dense, and the sampling strategy was specifically designed to best capture the boundary whose existence became apparent early on in the project. To achieve this we carefully studied maps and geological

reports to locate lamprophyres in key localities that had not been studied before (e.g., samples HAL, TAV, CRO, HOR, BUC).

In the ms we refer to the study by Canning et al (1996), who used Nd isotopes of post-Caledonian lamprophyres to delineate terrane boundaries in the mantle, an approach that admittedly closely matches ours. Note, however, that Canning et al sampled across *already known* lithospheric-scale boundaries, whereas we have discovered a completely unknown and unexpected boundary, so the novelty also clearly lies in the result (admittedly, some serendipity was involved).

We have reined in our statements with respect to novelty somewhat by referring to a 'new, mantle-based perspective', so we hope that the new ms is acceptable for the reviewers, whilst still suitable for Nature Communications.

2. Can the authors demonstrate that this study has important implications that extend beyond SW England? In other words, can the authors demonstrate that these findings will be of wide interest to those who do not work on the geology of SW England? If so, then I think the MS could be 'better sold' to appeal to a wider audience.

REPLY: The main implication that can be 'exported' is the method/approach. We have already started to apply the approach to other parts of the Variscan belt, collating published isotopic data on minettes and collecting new ones where gaps exist. This will hopefully lead to further manuscripts on the subject in the future. This can also be applied to other mountain belts, as post-orogenic lamprophyres are reported for many orogens around the world.

3. Is there any seismic evidence to support the idea that a steep-angled terrane boundary exists beneath the Culm Basin?

REPLY: We would clearly welcome studies that would either re-evaluate existing geophysical data or collect new data. The SWAT 'deep' seismic profiles collected in the region in the mid-80's by the BIRPS/ECORS consortium only focussed on the crust and the uppermost part of the lithospheric mantle. Sibuet et al. (JGR, 1990) have highlighted an area with several prominent and somewhat anomalous 'Caledonian' (i.e., north-dipping deep crustal - upper mantle reflectors) structures in SWAT3 profile, in the area of our proposed terrane boundary. While this is very enticing, the evidence isn't probably strong enough yet to put it forward in our manuscript as support for our model. Better deep lithospheric seismic (or magneto-tellurics?) data would obviously provide the ultimate test for our model.

Although points one and two will ultimately depend on the Editor's discretion, I would encourage giving the authors an opportunity to address them.

In addition to the points above, I have included a number of more minor comments/suggestions below:

Lines 6-8: See point 1 above. Have the authors overstated how novel the approach is given the use of similar techniques to map out the Caledonian

terranes of Scotland? Again, I should stress that I have no reservations about the novelty of the findings.

REPLY: OK, see reply below (comment referring to line 60 in original ms).

Line 11: I think it would be fairer to stress that this is a 'supposed' lithospheric discontinuity.

REPLY: Agreed, we have added 'postulated'.

Line 15: "subtle surface expression" seems rather vague. Is it simply the case that there is no surface expression?

REPLY: Agreed, but the phrase was deleted anyway in our attempt to reduce the abstract length to 150 words.

Line 34: that instead of which

REPLY: Happy to oblige, even if this is a somewhat personal preference (to us, 'which' as a pronoun starting a restrictive clause (i.e., without a preceding comma) has the same meaning as 'that').

Line 36: change the expression "their wandering years"

REPLY: OK, to us it seemed like a permissible style-figure in an introduction, but it has been removed (replaced by 'movements').

Line 37: Give the age of the Variscan Orogeny for less familiar readers

REPLY: OK, commonly accepted age of completion of the Variscan Orogeny added as "Late Carboniferous".

Line 50: Are most ophiolites not considered rather anomalous (tectonically and compositionally) by comparison to standard MORs?

REPLY: That is again a fair comment. We note, however, this discussion is generally with respect to the supra-subduction zone origin of the vast majority of ophiolites. Such a SSZ origin is completely lacking for the Lizard Ophiolite thus far, and the Lizard seems to be one of the few MORB-type ophiolites in the world! This seems a comment though rather than a suggested change... no changes were made here.

Line 52: Could metasomatism have occurred even earlier?

REPLY: That is indeed one of the conclusions of the ms!

Line 60: See previous comment about how novel the method is.

REPLY: We refer to our reply above, under major point 1.

Line 84-87: Are you referring to published data about the similarly aged potassic lavas? If so, a reference is needed.

REPLY: Reference to Thorpe et al. (1987) has been added.

Line 94-97: I would include the names of the labs used here. I appreciate that this is included in the supplementary information.

REPLY: Agreed. We have added a short methods section with a bit more information. The main method section is still in the supplementary file.

Line 103 – What exactly is meant by ‘high alkali and volatile contents’

REPLY: This statement is about lamprophyres in general, and there is quite a bit of variability in terms of absolute numbers. We think it is acceptable to be qualitative here, since the relevant rocks from the study area are discussed in fully quantitative terms.

Line 119 – reference needed for 60 km depth comment

REPLY: Garnet is stable in fertile mantle rocks (lherzolites) near the solidus at $P > 2.5$ GPa (as shown in many undergraduate textbooks, going back to O’Hara 1971 and Green & Ringwood, 1967). We have added a reference to the review paper by Wood et al. (2013) in Elements, because they also specifically mention and explain the trace element ‘garnet signature’. Because they specifically give a value of 85 km for the garnet signature, we have changed the minimum depth to 60-85 km.

Line 120 – reference for lithospheric thickness in this region?

REPLY: We have included a reference to Artemieva’s global TC1 model, which gives c. 100-125 km for the *present-day* lithospheric thickness, assuming that that value is a lower estimate for a lithospheric thickness at the end of the Variscan Orogeny at the start of the post-orogenic extension phase, but prior to the opening of the Atlantic and the extension forming the English Channel.

Line 134 – I’m not really sure what the term ‘melting anomaly’ refers to. Can this be changed/clarified?

REPLY: OK, title of the section is changed to ‘Mapping the Depth and Degree of Post-Orogenic Mantle Melting’ so the meaning should be more self-evident.

Line 135 – proxy rather than measure

REPLY: OK, replacement made.

Line 148 – other studies use similar trace element ratios to identify garnet signatures. Please reference

REPLY: Indeed, this is a very standard approach in igneous petrology. Reference to Wood et al. (2013) inserted again.

Line 150 – again, please clarify what you mean by ‘melting anomaly’

REPLY: OK, text changed to make it clearer. Replaced by “The mantle domain that experienced the shallowest and highest degree of post-orogenic, Early Permian mantle melting thus mapped out (figure 3a and b) underlies a region of Carboniferous sedimentation (the Culm basin)”.

Line 160 – The argument for these samples not being altered could do with strengthening. Simply saying that the samples look fresh is not sufficient. Can some form of cryptic alteration be ruled out?

REPLY: this topic is addressed in detail further below (it was also the main comment of reviewer 2).

Line 162 – use colon after ‘well’

REPLY: OK, colon added.

Line 172 – why can’t metasomatism have occurred much earlier?

REPLY: We fully agree, and that is indeed our conclusion (we say that it may be Cadomian in origin later). At this point we just report the conclusions of four other studies, in which the radiogenic Sr is interpreted in a subduction framework immediately before or at the same time as lamprophyre emplacement. We have made a minor change to ‘signpost’ our interpretation here already.

Line 181 – change ‘is’ to ‘provides’

REPLY: OK, replacement made.

Lines 191-198 – there are some interesting points here that could be further developed (although I accept that space is limited). The absence of any unconformity within the Devonian sequence of SW England is in marked contrast to the Devonian successions further north in Wales and northern England which are punctuated by the c. 400 Ma Acadian unconformity. SW England therefore appears to have escaped Acadian deformation. Woodcock et al (2007) highlighted these issues and suggested that SW England may have been juxtaposed with the Acadian belt along the Bristol Channel Fault Zone during the latter part of the Variscan. These authors favour a model where SW England was located adjacent to the Brabant Massif (Belgium) during the Devonian, thus allowing it to escape Acadian deformation. Interestingly, an earlier (early Devonian) deformation even (the Brabantian) is recognised in the Brabantian Massif that pre-dates the Devonian successions of SW England. Might the amalgamation of the two terranes proposed in this MS be related to the Brabantian event if the model of Woodcock et al (2007) is accepted?

REPLY: We appreciate this detailed and very interesting comment by the reviewer. It is clear that the timing of the terrane juxtaposition is as yet not fully constrained. If, following Woodcock et al., we accept the view that there is no significant unconformity in the Lower Devonian sedimentary successions in southwest Britain (and therefore, that they were not affected by the Acadian event), then the terrane boundary is older. We note that recent work by the reviewer (Miles & Woodcock, Lithos 2018) on the geochronology of the Shap granite dates the culmination of the Acadian event at 405 Ma, so it seems to possibly older than the ~395-400 Ma Brabantian event (Sinbutin et al., 2009). So while it does seem an attractive proposition to link the terrane amalgamation to either the Acadian or Brabantian phase, that remains difficult to reconcile with the absence of an unconformity around Emsian-Eifelian-Givetian times in the corresponding part of the sedimentary successions of southwest Britain (e.g., Meadfoot Group, Staddon grit), and the boundary is probably older.

Line 227 – same question about how novel the method is.

REPLY: OK, the word ‘novel’ has been removed here.

Line 228: again, any seismic evidence to support the idea that a steep terrane boundary exists?

REPLY: (See longer reply above). We have found no strong reports so far, and we would clearly welcome studies that would either re-evaluate existing geophysical data or collect new data.

Line 239 – delete ‘already’

REPLY: OK, deleted.

Line 245-254 – Whilst I accept that it seems likely that mineralisation may be a characteristic of the Armorican terrane, the absence of any granites in the Avalonian terrane means that it’s potential for generating mineralised ore fields is limited.

REPLY: We stated in the conclusions: *“This shows that the Armorican lower crust generally had the right composition (e.g., metagreywacke; Simons et al., 2017) to produce the Sn-W-rich peraluminous granitic magmas, as opposed to the crust of the Avalonian terrane”*. So we fully agreed with the reviewer that the crustal melting producing the peraluminous granites is the key – the mineralization is more a consequence of the granitic magmatism. We don’t think that a change is needed here.

Figures – these seem fine. I would consider re-ordering some of the figures such that a map is presented early on.

REPLY: OK, that is easily arranged, by referring to the map early on, so that it becomes figure 1. This has been implemented in the revision. We have also clearly shown the Lizard (and Start) ophiolites on the map, an omission from the original ms.

I wish the authors all the best in addressing these comments and I look forward to reading their published work

Yours Sincerely
Andrew Miles
30th April 2018

Reviewer #2 (Remarks to the Author):

Mapping a hidden terrane boundary 1 in the mantle lithosphere with lamprophyres. Review by Brendan Murphy

A plethora of tectonic studies inevitably focus on the definition and timing of crustal boundaries (sutures) between terranes. But when two crustal fragments collide, the fate of their respective sub-continental lithospheric mantle (SCLM) is rarely considered. This is a well written manuscript that uses geochemical and isotopic tracking to deduce the location of boundary between (respectively) sub-continental lithospheric mantle that lies beneath Avalonia and Armorica (Cadomia).

Isotopic tracking the evolution of the SCLM has been done before. The innovation here is that the approach is employed to deduce the deep mantle expression of a terrane boundary and that lamprophyres are to compare the deep (garnet lherzolite) SCLM beneath both terranes.

The results would be of interest to the many geologists working in southern Britain, and more generally to a global set of geologist who are trying to employ innovative methods to deduce the existence of terrane boundaries.

I have made many comments directly on the annotated pdf (attached). Most of these comments deal with clarifying the text, or some judicious rephrasing rather than scientific issues. Accordingly, I think this manuscript is acceptable after minor revisions. The authors should evaluate (rather than accept) the suggested edits, because some of them may have inadvertently changed the meaning they intended.

My main concern is the rather cavalier way in which the potential effects of alteration are treated. I have seen these rocks in the field and in thin section and they ARE altered. This alteration is unlikely to significantly affect the more robust Sm-Nd isotopic systematics, but they are certainly likely to affect the more fickle Rb-Sr isotopic systematics. The differences in Sr isotopic signature seem more profound than Nd. Therefore the author(s) should justify their assumption about how representative the Sr isotopic analyses are of the original magma composition.

REPLY: While it is undeniable that there are plenty of altered lamprophyres in the study area (and these may be the ones the reviewer is referring to), there are also several fresh occurrences, as shown by photos B, E and F in the supplementary information. We cannot use Loss On Ignition as a measure of alteration like in many other igneous rocks, but this is clearly not an option in the lamprophyres, which have a high abundance of primary micas and carbonate. Hall (Mineralogical Magazine, 1982) provided a detailed mineralogical description of the Pendennis lamprophyre (PEN in our study as) writing "*There is no petrographic evidence of secondary alteration; the country rocks are almost devoid of carbonates; there are no carbonate veins in either the minette or the*

country rocks; and the dolomite of the minette is zoned in the same way as the mica”.

In our sampling, we have made great effort to sample the freshest rocks available in the exposure, and no strongly altered lamprophyres were analysed for their isotope systematics. Thin sections were made for each sample. Qualitative petrographic criteria used for relative ‘freshness’ that we employed are: a lack of cloudiness in the feldspars (no saussurite, sericite, or fine-grained chlorite); presence of primary carbonate (in particular when associated with apatite) in triple junctions; and presence of zoned micas (pale brown phlogopitic cores with brown pleochroic biotitic rims) without the alteration to opaque phases often seen in altered dark micas. However, even the freshest samples contain altered olivine, where the olivine is generally replaced by secondary carbonate and opaque phases, and we attribute this to autometasomatism due to the H₂O- and CO₂-rich nature of the magma (Rock, 1987). So all samples are altered to some extent, but in several samples (KIL, KHQ, LEM, PEN, MAW) the petrographic evidence for alteration is so minor and mostly of autometasomatic nature (which would not affect the isotope chemistry) that they are considered ‘fresh’.

We also agree with the reviewer that Nd isotopes are unlikely to be modified even in significantly altered samples, due to the very low concentrations of Nd in plausible fluids causing alteration (~3 ppt in seawater, Dickin 1995 textbook) and the low mobility of REE. The Sr isotopes are in general more susceptible to alteration because of the higher solubility of Sr (c. 8 ppm in seawater), but the very high concentrations of Sr in the lamprophyres before any alteration (e.g., ~1600 ppm in the very fresh Pendennis sample) make them also much more resistant to modification of their Sr isotopic ratios than most other igneous rocks.

Interestingly, in the northern terrane, ALL the analysed lamprophyres, even the moderately altered ones (THO, HAL), plot on the 290 Ma mantle array. This effectively rules out significant modification of the Sr isotopic ratio to more radiogenic ‘crustal’ values as a result of secondary alteration in our sample set. The only two samples in the northern terrane that plot significantly off the mantle array are both altered K-rich basalts: one is shifted to higher, more radiogenic Sr values (POS), and one to seemingly less radiogenic values (KNO). These basaltic rocks had much lower Sr concentrations to start with, and were therefore more prone to partial isotopic resetting than lamprophyres. The conclusion must therefore be that the Sr and Nd isotopic difference between the two groups is a primary, magmatic feature, and not the result of secondary alteration.

REVIEWER 2 COMMENTS PROVIDED AS ANNOTATIONS ON MS ARE COPIED
HERE

Line 32: Given that the type area of Avalonia is in Atlantic Canada, these list of references are very biased towards Europe, which are interpreted correlatives. Suggest add a few North American references

REPLY: OK, fair comment. References to Van Staal et al (2009) and (2012) and Murphy et al (2010) have been added.

Line 38: No, it is NOT a type locality. That is in Atlantic Canada

REPLY: OK, comment accepted. "One of the type localities..." has been replaced by "A key locality..."

Line 49 (in relation to a reference to a paper by Murphy et al. (2010): A better reference (which includes reference to Sm-Nd of Lizard is:

Murphy, J.B., Cousens, B.L., Braid, J.A., Strachan, R.A., Dostal, J., Keppie, J.D., and Nance, R.D., 2011. Highly depleted oceanic lithosphere in the Rheic Ocean: implications for Paleozoic reconstructions. *Lithos*, 123, 165-175.

REPLY: OK, happy to replace Murphy et al (2010) by Murphy et al. (2011) as suggested

Line 56: See Arenas et al. 2014

Arenas, R., Díez Fernández, R. Sánchez Martínez, S., Gerdes, A., Fernández-Suárez J., Albert, R.. 2014. Two-stage collision: Exploring the birth of Pangea in the Variscan terranes. *Gondwana Research* 25, 756-763.

REPLY: Thanks, that is a useful reference to add

Line 58-59: This is a very loose sentence. First, I think there are several models and second you need some references

REPLY: Fair comment. This section has been reshuffled so it sounds more 'logical' and the sentence in question has been removed.

Line 62: The use of mafic rocks is not that novel. See:

Murphy, J.B., and Dostal, J., 2007, Continental mafic magmatism of different ages in the same terrane: constraints on the evolution of an enriched mantle source. *Geology*, 35, 335-338.

REPLY: We are well aware of this paper, and we think it is an excellent and seminal study showing how to use (isotope) geochemistry of mafic rocks to trace the *temporal* evolution of the mantle lithospheric part of a terrane. The paper isn't about *mapping* terrane boundaries and is less relevant as a citation in this context. No modifications have been made.

Line 111: what do you mean by SiO₂ rich, specify typical range

REPLY: Experimental liquids had 52.1-63.5 wt% SiO₂. This range has been inserted in the text.

Figure 2: Caption needs to explain initial

REPLY: OK, added.

Figure 2: This difference really depends on how representative the $^{87}\text{Sr}/^{86}\text{Sr}$ values are

REPLY: We are not certain what is meant here... If the word 'representative' refers to possibly modification due to alteration, we would like to refer to our reply above, where we have addressed/discussed this issue.

Figure 2 (in reference to Lizard dyke values): refer to table where these data can be found

REPLY: OK, this was a dyke analyzed alongside the lamprophyres. The values have been added to the data table.

Figure 3c: depends on reliability of alkalis

REPLY: That is true, and it is likely that the more strongly altered samples could have lost some alkalis by leaching. The strong correlation with Rare Earth Elements (e.g., Dy/Yb) and the strongly clustered geographical distribution makes it more likely that the primary control is source melting rather than alteration.

Figure 4: clearly label terrane boundary in mantle and crust

REPLY: OK, labeling made clearer

Unclear what you mean by "earlier". Pre-Variscan? If so, were they once side by side on the Gondwanan margin

REPLY: Because this metasomatism affected both terranes, it must have happened syn- or post- terrane juxtaposition, and we referred to it as a kind of 'overlap assemblage'. We wrote in the text: "The more widespread potassic-hydrous metasomatism that overprinted the terrane boundary can most easily be explained as having occurred above a north-dipping slab during Variscan subduction of oceanic lithosphere, although Late Devonian-Early Carboniferous alkaline intra-plate magmatism in the region (Floyd, 1992) is not fully discounted here as a contributing cause of the metasomatism." We would like to leave it in the absence of further constraints.

Figure 5: clearly label the suture across Cornwall.

REPLY: OK, suture is shown more clearly, and as a continuous line

Why do you assume the crustal suture and the mantle suture are the same line? I would have thought that to be very unlikely

REPLY: In our ms we have located the boundary in the base of the mantle lithosphere. We have no new constraints on the location in the lower crust. However, the observation that the boundary also essentially marks the northern limit of granite emplacement with the associated W-Sn mineralization (figure 6) does indeed suggest that the mantle and lower crustal boundaries could coincide. We are not sure why that would be 'unlikely', especially if the boundary is of a transcurrent nature?

Table I: Give GPS Locations of all samples

REPLY: They have all been given (already in the original ms), with a sort phrase to describe the sample site, in SI-table I.